# The RNA-binding protein HuR is a negative regulator in adipogenesis

Diana Teh Chee Siang[1,4], Yen Ching Lim[1,4], Aung Maung Maung Kyaw[1], Khaing Nwe Win[1], Sook Yoong Chia[1], Ufuk Degirmenci[1], Xiang Hu[2], Bryan C. Tan[1], Arcinas Camille Esther Walet [1], Lei Sun [1,3]* & Dan Xu [1]*

Human antigen R (HuR) is an essential regulator of RNA metabolism, but its function in metabolism remains unclear. This study identifies HuR as a major repressor during adipogenesis. Knockdown and overexpression of HuR in primary adipocyte culture enhances and inhibits adipogenesis in vitro, respectively. Fat-specific knockout of HuR significantly enhances adipogenic gene program in adipose tissues, accompanied by a systemic glucose intolerance and insulin resistance. HuR knockout also results in depot-specific phenotypes: it can repress myogenesis program in brown fat, enhance inflammation program in epidydimal white fat and induce browning program in inguinal white fat. Mechanistically, HuR may inhibit adipogenesis by recognizing and modulating the stability of hundreds of adipocyte transcripts including Insig1, a negative regulator during adipogenesis. Taken together, our work establishes HuR as an important posttranscriptional regulator of adipogenesis and provides insights into how RNA processing contributes to adipocyte development.

[1] Cardiovascular and Metabolic Disorders Program, Duke-NUS Medical School, 8 College Road, Singapore 169857, Singapore. [2] Departments of Endocrine and Metabolic Diseases, The first Affiliated Hospital of Wenzhou Medical University, Wenzhou, Zhejiang 325035, China. [3] Institute of Molecular and Cell Biology, 61 Biopolis Drive, Proteos, Singapore 138673, Singapore. [4] These authors contributed equally: Diana Teh Chee Siang, Yen Ching Lim. *email: sun.lei@duke-nus.edu.sg; dan.xu@duke-nus.edu.sg

Obesity is characterized by excessive lipid storage in adipose tissue, leading to an increase of metabolic disorders such as type 2 diabetes, cardiovascular disease and some types of cancer. Adipose tissue expansion is a complex process involving an enlargement of existing adipocytes (hypertrophy) and an increased number of adipocytes (hyperplasia) through adipogenesis[1]. A better understanding of the molecular mechanism underlying adipogenesis is necessary for developing the treatment of obesity and its associated diseases. Earlier studies have identified a variety of transcriptional factors that govern different aspects of adipogenesis by regulating downstream genes at the transcriptional level[2–4]. However, the regulation of adipogenesis at the posttranscriptional level has yet to be fully understood.

RNA-binding proteins (RBPs) play an essential role in governing the fate of mRNA transcripts from biogenesis, stabilization, translation to RNA decay[5,6]. Several RBPs have been reported as adipocyte regulators by affecting different aspects of RNA processing. SFRS10 and Sam 68 modulate the adipocyte development and lipid metabolism by controlling alternative splicing[7–9]. KSRP (KH-type splicing regulatory protein) ablation promotes browning of white adipose tissues by reducing miR-150 expression[10]. RNA-binding protein IMP2 controls energy metabolism by inhibiting the translation efficiency of UCP1 and other mitochondrial mRNAs[11]. PSPC1 (paraspeckle component 1) promotes adipogenesis by facilitating the export of many adipocyte RNAs, including the RNA encoding the transcriptional regulator EBF1, from the nucleus to the cytosol[12]. Our recent work indicates that Ybx2 is an important regulator that controls brown adipose tissue activation by regulating mRNA stability[13]. However, the functions of most RBPs in adipocytes remain unexplored.

HuR (also known as Elavl1) is a universally expressed RBP. It recognizes and binds to the AU-rich elements (ARE) in 3′UTR of its target mRNAs, and enhances mRNA stability and translation often by competing for AREs occupancy against mRNA-destabilizing modulators[14–16]. HuR plays a crucial role in the development and functions of several cell types. HuR homozygous knockout mice are lethal[17]. Tissue-specific knockout models have demonstrated that HuR is essential for hematopoietic progenitor cell survival, T cell selection and chemotaxis, and B cell antibody response[18–20]. HuR confines the pro-inflammatory cytokine production in myeloid cells and protects the mice from colorectal carcinogenesis[21]. HuR plays a neuroprotective role in the brain and regulates small intestinal epithelial homeostasis and proliferation[22,23]. The depletion of HuR results in an inhibition of C/EBPβ protein expression and an attenuation of the 3T3-L1 cells differentiation process[24,25]. Despite several decades of study on HuR, our knowledge about its functions and mechanisms is still limited to a small number of cell types. Particularly, we know little about the physiological role of HuR in any major metabolic tissues.

Here, we show that HuR is a key repressor for adipogenesis in both white adipose tissue (WAT) and brown adipose tissue (BAT). Loss-of-HuR in adipose tissue significantly increase fat mass in mice, accompanied with glucose intolerance and insulin resistance. HuR recognizes a set of mRNAs, including insulin-induced gene 1 (Insig1), which is destabilized and downregulated in the absence of HuR. Our study establishes that HuR is an important posttranscriptional regulator of adipogenesis and provides insights into how posttranscriptional processes contribute to adipocyte development.

## Results

**HuR is a repressor for adipocyte differentiation in vitro**. To explore the potential function of HuR in adipocytes, we first examined HuR expression during adipocyte differentiation in vitro. We isolated the SVF (stroma-vascular fraction) and mature adipocytes from Brown adipose tissue (BAT), inguinal white adipose tissue (iWAT) and epididymal white adipose tissue (eWAT) of 2-month-old C57BL6 mice. Western blot analysis revealed that expression of HuR was decreased in mature adipocytes of three different adipose tissues (Fig. 1a). Consistent with it, HuR was also downregulated during the lipid accumulation phase (day 4 to day 8) in an in vitro differentiation time course of primary adipocytes (Fig. 1b). These data are suggestive of a functional link between HuR and adipogenesis.

To investigate the function of HuR in adipogenesis, we depleted HuR by infecting brown and white preadipocytes with retroviruses expressing targeting shRNAs and then induced cells to differentiate for 5 days. Knocking down HuR in brown adipocytes significantly promoted brown fat marker and pan-adipocyte marker expression detected by real-time PCR (Fig. 1c) and lipid accumulation assessed by Oil Red O staining (Fig. 1d). Similar results were obtained in WAT (Fig. 1e, f). Conversely, we overexpressed HuR in primary brown and white preadipocytes using retrovirus, followed by induction of differentiation. Overexpression of HuR greatly repressed lipid accumulation (Fig. 1h, j), BAT-selective markers and pan-adipocyte markers expression (Fig. 1g, i). These data indicate that HuR plays a repressive role in adipogenesis.

To assess the function of HuR in human adipocytes, we overexpressed human HuR in primary preadipocytes isolated from human fetal interscapular BAT[26] and subcutaneous WAT. Similar as found in mouse adipocytes, overexpression of HuR severely inhibited lipid accumulation and marker expression in both human BAT and WAT adipocyte culture (Fig. 1k–n and Supplementary Fig. 1a). To test whether the function of HuR in human and mouse is interchangeable, we overexpressed human HuR in mouse BAT and WAT adipocyte culture. Interestingly, human HuR overexpression inhibited the differentiation of mouse BAT and WAT adipocytes (Supplementary Fig. 1b–d). Taken together, HuR is a repressor for both mouse and human adipogenesis in vitro.

**HuR knockout in adipose tissue results in increased fat mass**. To determine the function of HuR in vivo, we generated mice with a fat-specific deletion of the HuR gene by crossing *HuR^flox/flox* and *Adipoq*-Cre mice, referred to as HuR-FKO mice. First, we confirmed the lack of HuR in three different adipose tissues compared with control littermates (*HuR^flox/flox*; Cre-) by Real-time PCR (Supplementary Fig. 2a) and western blot (Fig. 2a). Knockout animals did not exhibit significant alteration in their body weights (Fig. 2b), body lengths (Supplementary Fig. 2b), food intake (Supplementary Fig. 2c) and energy expenditure (Supplementary Fig. 2d, e). However, EcoMRI scans indicated that body fat mass was clearly increased in FKO mice (Fig. 2c), which is mainly due to a significantly enlarged epididymal white depot (Fig. 2d, e). The inguinal white adipose and brown adipose of FKO were marginally bigger but not significant (Fig. 2d, e). Consistently, H&E staining showed hypertrophic epididymal white fat depots in HuR-FKO mice (Fig. 2f).

To test whether HuR-FKO mice developed insulin resistance in adipose tissue, we examined the phosphorylation of AKT in eWAT after insulin administration and observed decreased phosphorylation of AKT in the HuR knockout eWAT (Fig. 2g). Since the adipose tissue insulin resistance is often associated with systemic insulin resistance, we conducted glucose tolerance test (GTT) and insulin tolerance test (ITT). HuR-FKO mice became whole-body glucose intolerant and displayed insulin resistance (Fig. 2h, i). Taken together, deficiency of HuR in adipose tissues results in an increased fat content and an impaired insulin sensitivity.

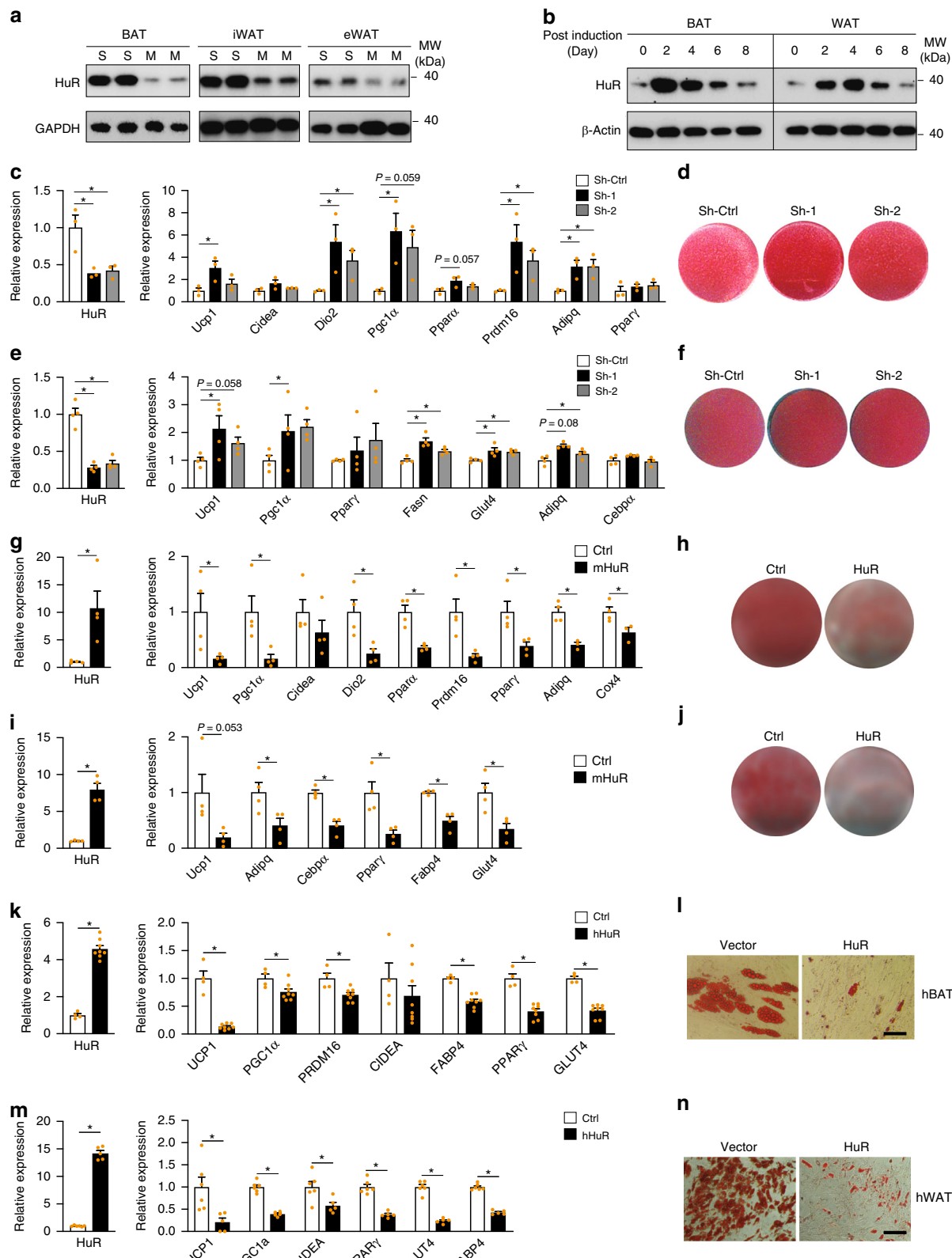

**HuR knockout in eWAT enhances adipogenesis and inflammation.** To further examine the effect of HuR on eWAT at the molecular level, we conducted RNA-seq of eWAT isolated from 3-month-old HuR-FKO and control littermates. Consistent with the increased epididymal fat mass, GSEA analysis revealed an enhanced adipogenesis pathway in eWAT of HuR-FKO mice (Fig. 3a, b). We confirmed these molecular changes by real-time

PCR to examine the expression of key marker genes in adipogenesis and lipid metabolism. Most of the examined genes are upregulated (Fig. 3c, d). Thus, HuR deficiency promotes the adipogenesis pathway at molecular levels.

In addition, we observed some upregulated immune response pathways, such as interferon gamma response and inflammatory response (Fig. 3a, e, f), which were successfully validated by

**Fig. 1 HuR represses adipocyte differentiation in vitro. a** Western blot to compare the HuR expression in Stromal vascular fraction of adipose tissue (S) and mature adipocytes (M). **b** Western blot to examine the protein levels of HuR in primary brown and white adipocyte differentiation time course. **c** Primary brown preadipocytes were infected by retroviral shRNAs targeting HuR followed by induction of differentiation for 5 days. Real-time PCR was used to measure the knockdown efficiency (left), BAT-selective marker and pan-adipocyte marker expression (right). $n = 3$. **d** Oil Red O staining to assess lipid accumulation. **e, f** Similar as in (**c, d**), but in primary white adipocyte culture. $n = 4$. **g** Retroviral overexpression of HuR in primary brown preadipocytes. Real-time PCR was conducted to examine HuR (left) and marker gene expression (right) of brown adipocyte culture expressing HuR or vector at day 5. $n = 4$ per group. **h** Oil Red O staining of HuR-overexpressing brown adipocytes. **I, j** Similar as in (**g, h**), but in primary white adipocyte culture. $n = 4$ per group. **k** Retroviral overexpression of HuR in human fetal brown adipocytes. Real-time PCR was used to determine the expression of HuR (left) and marker gene expression at day 21 post induction (Ctrl, $n = 4$; HuR, $n = 8$). **l** Oil red O staining for human brown adipocytes at day 21 post induction. The scale bar represents 100 μm. **m, n** similar as in (**k, l**) in Human subcutaneous WAT at day 14 post induction (Ctrl, $n = 6$; HuR, $n = 5$). The scale bars represent 100 μm. Error bars are mean ± SEM. Statistical significance using Student's $t$-test for (**g**), (**i**), (**k**), and (**m**), One-way ANOVA test for (**c**) and (**e**), *$p < 0.05$. Source data are provided as a Source Data file.

real-time PCR (Fig. 3g). Consistently, the IF staining of F4/80 clearly indicated that more macrophages were present in the HuR-FKO eWAT (Fig. 2f). It is possible that HuR knockout results in an enhanced inflammatory response in adipocytes per se or the increase in adipocyte size results in larger intercellular space which allows for the infiltration of immune cells. These two explanations are not mutually exclusive, and it is certainly possible that both effects contribute to the adipose tissue inflammation.

**HuR knockout in iWAT enhances adipogenesis and browning.** We determined the effect of HuR knockout on iWAT that is enriched for beige adipocytes. These beige adipocytes exhibit white adipocyte features at thermoneutrality but can undergo browning phenotype to take on BAT-like cellular and molecular phenotypes under environmental stimuli such as cold exposure[27]. We analyzed the global gene expression in iWAT samples from 3-month-old HuR-FKO and their control littermates by RNA-seq. Consistent with the findings in vitro, our GSEA analysis clearly demonstrated that the upregulated genes are enriched for adipogenesis and fatty acid metabolism pathways (Fig. 4a–c), which can be successfully confirmed by real-time PCR (Fig. 4e). Therefore, HuR knockout results in enhanced adipogenesis in both eWAT and iWAT.

Another enhanced pathway in HuR-FKO iWAT was oxidative phosphorylation (Fig. 4a, d). The brown fat markers such as Ucp1, Dio2, Fabp3, and Elvol3, and beige fat markers were among the most significantly upregulated genes in HuR-FKO iWAT (Fig. 4f). At the protein level, we confirmed the upregulation of UCP1 and CIDEA in HuR-FKO iWAT by western blot (Fig. 4g). Thus, lack of HuR results in increased adipogenesis and browning in iWAT.

HuR depletion resulted in browning in iWAT but inflammation in eWAT, suggesting that HuR may play different roles in distinct depots. To analyze the depot-specific effects of HuR knockout, we compared the altered pathways in these two depots. Most altered pathways displayed clear depot-specific pattern. For instance, HuR knockout enhanced oxidative phosphorylation in iWAT but not in eWAT, induced interferon responses, Tnfα signaling in eWAT but not iWAT, and repressed angiogenesis, hypoxia, and IL6/JAK/ STAT3 signaling in iWAT but not eWAT (Fig. 4h). However, the fatty acid metabolism and adipogenesis pathways were significantly upregulated in both depots. Thus, while HuR knockout has differentially affected a variety of pathways in iWAT vs. eWAT, it promotes adipogenesis in both depots.

**HuR knockout in BAT promotes brown adipogenesis.** We next investigated the role of HuR in brown adipocyte at the molecular level. We conducted RNA-seq for BAT from HuR-FKO and their control littermates. GSEA analysis revealed a downregulated

myogenesis pathway in HuR-FKO BAT (Fig. 5a). As brown adipocyte and skeletal muscle cells share the same precursor[28], the downregulation of muscle markers suggested an enhanced commitment to the brown fat lineages. Real-time PCR confirmed that pan-adipogenic markers such as Fabp4, and brown fat-selective markers such as Ucp1, Pgc1α, and Fabp3 were increased in HuR-FKO BAT (Fig. 5b). Conversely, the muscle markers are significantly downregulated in HuR-FKO BAT (Fig. 5c). Western blot was conducted to confirm the increased UCP1 and PGC1α but decreased DESMIN protein levels in HuR-FKO BAT (Fig. 5d). Thus, HuR-FKO BAT developed enhanced brown adipogenesis.

Since HuR was deleted in all adipose tissues in the FKO mice, we wonder whether the crosstalk between different adipose depots[29] might affect the observed BAT's phenotypes. To test this possibility, we generated $HuR^{flox/flox}$; $UCP1$-cre+ mice (henceforth designated as HuR-BATKO) to delete HuR specifically in BAT. Western blot was conducted to confirm the deletion efficiency (Fig. 5e). No significant differences in body weight, fat mass, and organ weights were observed (Supplementary Fig. 3a–c). Histology analysis of BAT clearly demonstrated that the brown adipocytes in HuR-BATKO mice were smaller (Fig. 5f) and brown fat markers were greatly increased in HuR-BATKO BAT (Fig. 5g). Taken together, these results indicate that deletion of HuR in BAT promotes brown adipogenesis in vivo. To test whether HuR deletion in BAT may functionally crosstalk to other fat depots, we examined the eWAT where HuR was not ablated. As expected, we did not observe any significant difference in fat organ weight and gene expression of most markers (Supplementary Fig. 3c, d). The GTT and ITT assay exhibited little change between the control and HuR-BATKO mice (Supplementary Fig. 3e, f). Thus, the functional influence of HuR knockout in BAT is not sufficient to alter systemic metabolic homeostasis.

**HuR targets and stabilizes Insig1 mRNA.** To determine the mechanism underlying HuR's function during adipogenesis, we comprehensively identified the mRNA targets of HuR by conducting RNA immunoprecipitation followed by RNA-seq (RIP-Seq) in eWAT samples. The successful HuR precipitation by HuR antibody but not IgG was confirmed by western blot (Fig. 6a). We selected 200 genes most enriched by HuR immunoprecipitation as its targets (Supplementary Data 4). Consistent with earlier reports[14,16], the targets of HuR were highly enriched for RNA metabolism processing (Fig. 6b). As expected, most of HuR's target mRNAs, predicted by AREsite2[30], bore ARE signature (Supplementary Data 5).

To determine the effect of HuR deficiency on its targets, we performed RNA-seq to examine the expression change of these target mRNAs. We found that the HuR-targeted mRNAs, compared with other transcripts, tended to be more downregulated in the HuR knockout samples (Fig. 6c), supporting a role of HuR in

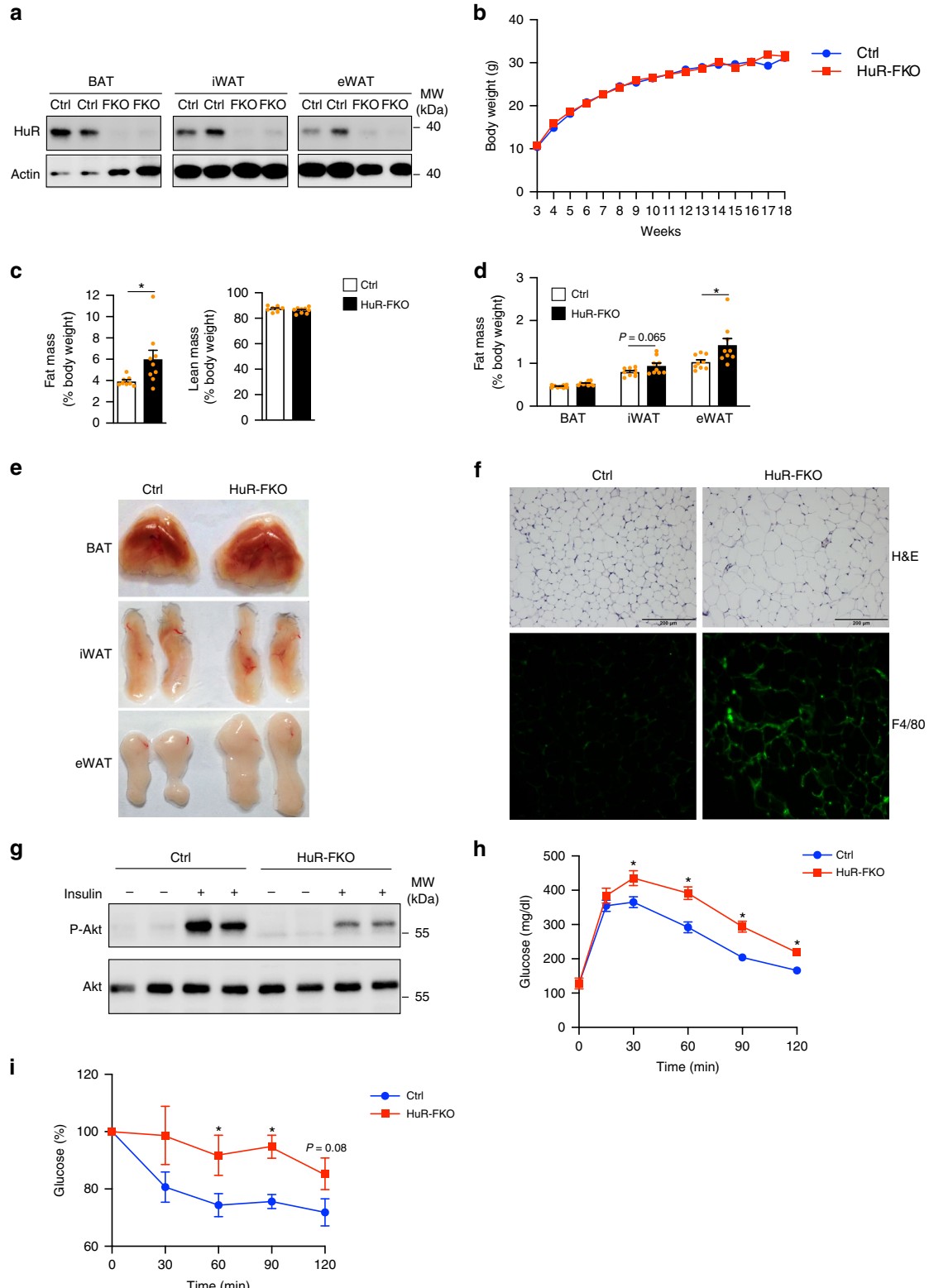

**Fig. 2 HuR knockout in adipose tissue results in increased fat mass. a** Western blot analysis of HuR expression in BAT, iWAT, and eWAT of HuR-KFO and control mice. **b** Body weight of male HuR-FKO and control (*HuR$^{flox/flox}$*) mice (Ctrl, $n = 7$; HuR-FKO, $n = 10$). **c** In vivo fat and lean mass by EcoMRI in 3-month-old male HuR-FKO and control littermates. (Ctrl, $n = 8$; HuR-FKO, $n = 9$). **d** The organ weights of BAT, iWAT, and eWAT in 3-month-old HuR-FKO and control littermates were normalized as a percentage of total body weight. $n = 9$ per group. **e** Representative picture of BAT, iWAT, and eWAT. **f** Morphological characteristics of eWAT by H&E staining and immunostaining of eWAT with F4/80. Scale bars represent 200 μM. **g** Insulin-induced AKT phosphorylation in eWAT from HuR-FKO and Ctrl mice. Five minutes after IP insulin injection, mice were then sacrificed to harvest eWAT and examined its phosphorylation levels of AKT by western blot. **h** Blood glucose levels during glucose tolerance test (GTT) ($n = 9$ per group) and **i** insulin tolerance test (ITT) of HuR-FKO and control littermates (Ctrl, $n = 8$; HuR-FKO, $n = 6$). Error bars are mean ± SEM. Statistical significance was determined by Student's *t*-test; *$p < 0.05$. Source data are provided as a Source Data file.

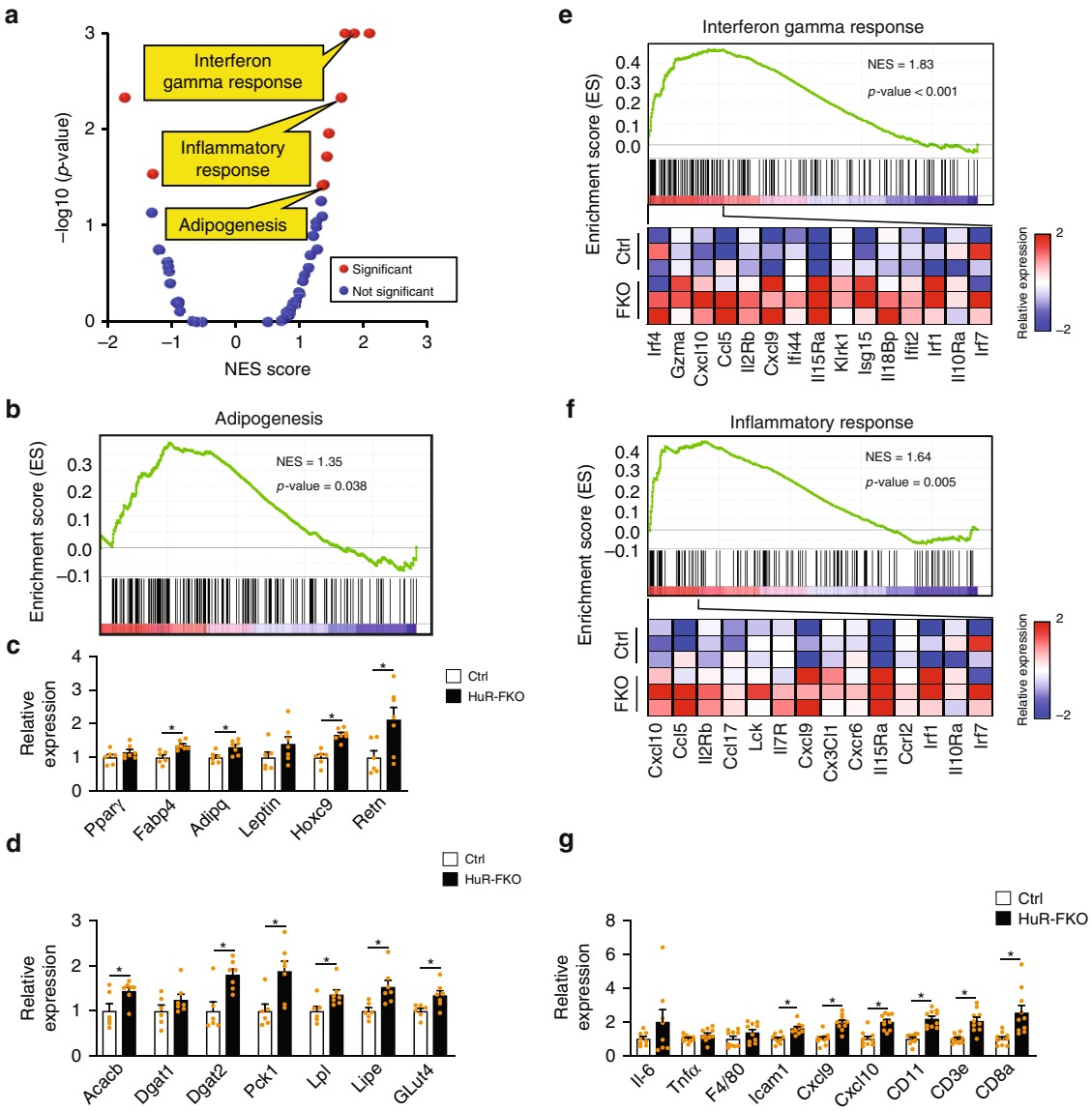

**Fig. 3 HuR knockout enhances adipogenesis and inflammation in eWAT. a** GSEA was performed to analyze the pathways affected by HuR depletion in eWAT. Scatterplot depicts the relationship between the *p*-value and normalized enrichment score (NES) of 50 "Hallmark" gene sets in MSigDB. Significant biological pathways are labeled in red. **b** GSEA in adipogenesis pathway. **c** Real-time PCR to confirm the expression of genes for white fat and adipogenesis (Ctrl, *n* = 6; HuR-FKO, *n* = 7). **d** Real-time PCR to confirm the expression of genes for lipogenesis and glucose uptake (Ctrl, *n* = 6; HuR-FKO, *n* = 7). **e** GSEA in interferon gamma response and **f** inflammatory response pathway. At the bottom of each panel shows a heatmap of representative genes found within the leading edge-subset of the biological pathway. The color intensity represents median centered gene expression (FPKM) with red and blue representing highly and lowly expressed genes, respectively. **g** Real-time PCR to confirm the expression of genes involved in inflammation. *n* = 10 per group. Error bars are mean ± SEM. Statistical significance was determined by Student's *t*-test; *$p < 0.05$. Source data are provided as a Source Data file.

stabilizing its target mRNAs. As expected, HuR targets identified in our RIP-seq analysis showed little overlapping with the genes in the upregulated GSEA pathways in Fig. 3 (Supplementary Fig. 4a–i), which suggests that the upregulated inflammatory pathways in Fig. 3 are likely due to a downstream effect from HuR's direct targets.

To examine whether HuR may influence its targets at the translational level, we conducted ribosome profiling and quantified the ribosome protected fragment (RPF) in control and knockout eWAT (Supplementary Data 2). Quality control analysis indicated a tight correlation between sample replicates (Supplementary Fig. 5a, b) and significant RPF accumulation near the starting codon due to the pauses of the ribosome at these positions (Supplementary Fig. 5c, d). We then assessed the influence of HuR on translational efficiency calculated by the ratio between RPF and

mRNA levels, but did not observe any significant alteration (Supplementary Fig. 5e). Thus, HuR may not have a major impact on the translation of its target mRNAs.

Among the top HuR targets, Insig1 mRNA was significantly depleted in three adipose tissues of HuR-FKO mice by real-time PCR (Fig. 6d) and western blot (Supplementary Fig. 6a). Previous studies have shown that Insig1 repressed adipogenesis in 3T3-L1 cells and mouse model[31,32]. Indeed, overexpression of Insig1 repressed both primary BAT and WAT cells differentiation in vitro (Supplementary Fig. 6b, c), which confirmed the repressive role of Insig1 in adipogenesis. Thus, we hypothesized that HuR may repress adipogenesis at least partially through Insig1. To test this hypothesis, we first confirmed the binding between Insig1 mRNA and HuR by performing the RIP-PCR in tissue lysates from HuR-FKO and control eWAT. We observed a

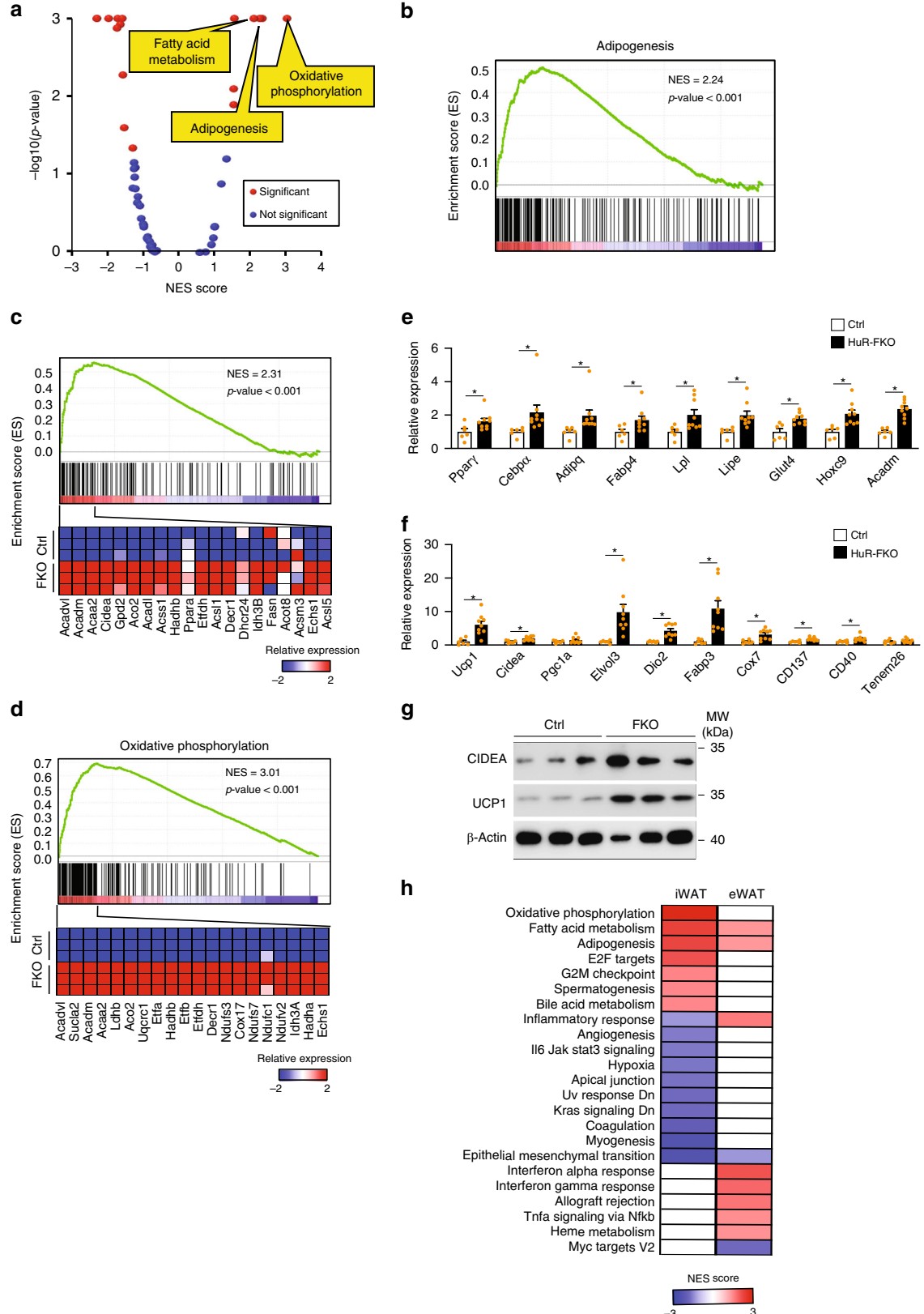

**Fig. 4 HuR knockout enhances adipogenesis and browning in iWAT. a** GSEA was performed to analyze the pathways affected by HuR depletion in iWAT. Scatterplot depicts the relationship between the *p*-value and NES score of 50 "Hallmark" gene sets in MSigDB. **b** GSEA on the biological pathway of adipogenesis, **c** fatty acid metabolism, and **d** oxidative phosphorylation. **e** Real-time PCR to confirm the expression of genes for adipogenesis, lipogenesis, and **f** BAT-selective and beige fat markers (Ctrl, *n* = 6; HuR-FKO, *n* = 9). **g** Western blot was performed to examine the protein level of Cidea and Ucp1 in iWAT from HuR-FKO and Ctrl mice. **h** The regulated pathways in HuR-FKO iWAT and eWAT are displayed in a heatmap. The color represents the NES for each pathway. Error bars are mean ± SEM. Statistical significance was determined by Student's *t*-test; *p < 0.05.

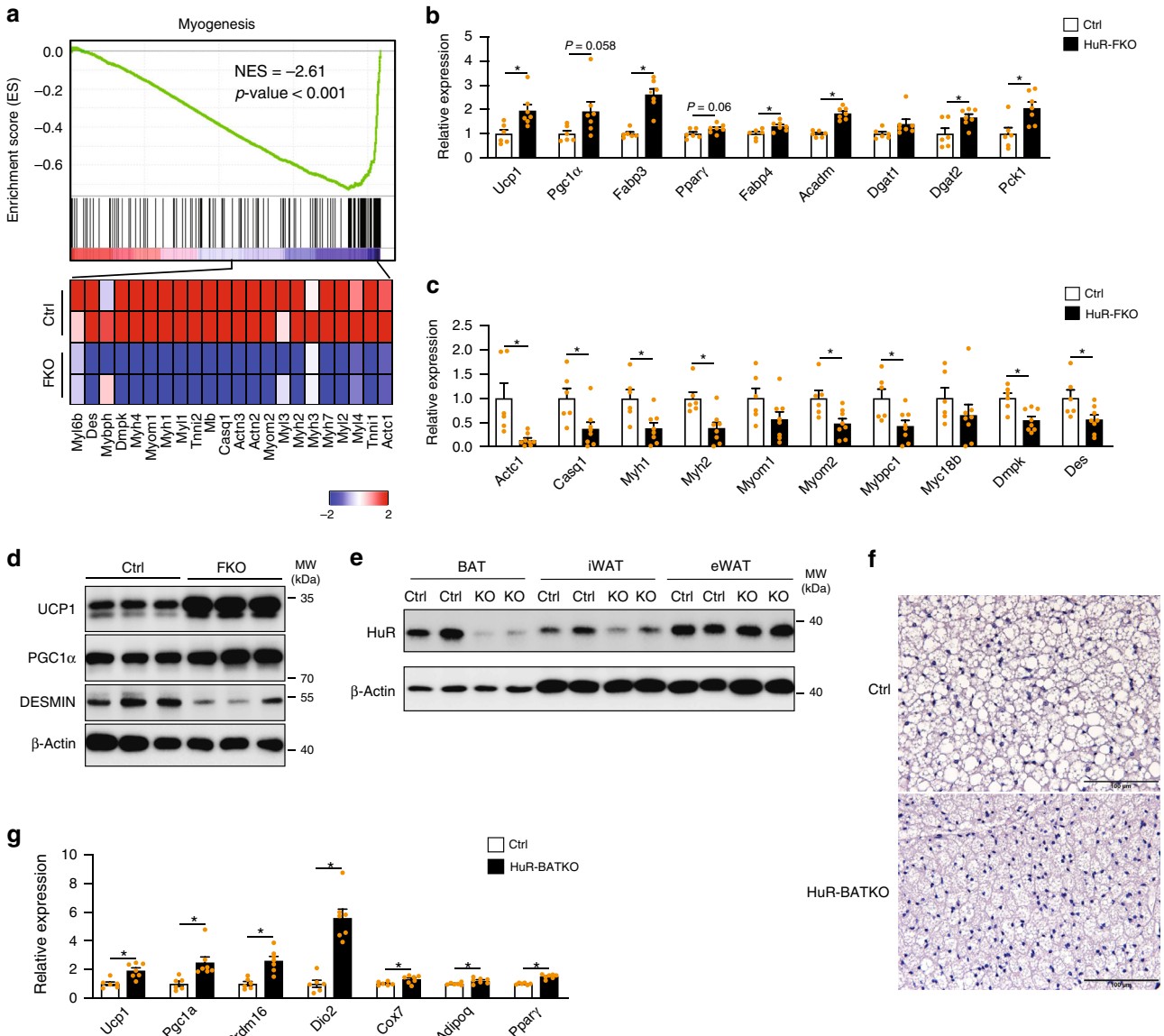

**Fig. 5 Deletion of HuR in BAT promotes brown adipogenesis. a** GSEA on HuR-FKO and Ctrl BAT indicated that myogenesis pathway was significantly downregulated in HuR knockout BAT. **b** Real-time PCR to confirm the expression of BAT-selective markers, adipogenesis, lipogenesis (Ctrl, $n = 6$; HuR-FKO, $n = 7$), and **c** muscle markers (Ctrl, $n = 6$; HuR-FKO, $n = 8$). **d** Western blot to exam the protein level of Ucp1, Pgc1a, and Desmin in BAT from HuR-FKO and Ctrl mice. **e** Western blot analysis of HuR expression in BAT, iWAT, and eWAT of HuR-BATKO and control mice. **f** Morphological characteristics of HuR-BATKO and control littermates by H&E staining. Scale bars represent 100 μM. **g** Real-time PCR to confirm the expression of genes for BAT-selective markers and adipogenesis in BAT from HuR-BATKO and Ctrl mice (Ctrl, $n = 6$; HuR-BATKO, $n = 7$). Error bars are mean ± SEM. Statistical significance was determined by Student's *t*-test; *$p < 0.05$. Source data are provided as a Source Data file.

significant reduction of Insig1 mRNA retrieved by HuR antibody in HuR-FKO in comparison with the control eWAT, but did not observe such a reduction for Fabp4 mRNA that did not bear putative HuR binding sites (Fig. 6e). To further identify HuR-binding sites in Insig1 3′UTR, we in vitro transcribed two Uridine-enriched RNA fragments from Insig1 3′UTR, which were predicted as HuR binding sites, and conducted RNA pull-down assay. HuR was successfully retrieved by both fragments (Fig. 6f). Together, these data demonstrate a molecular interaction between HuR and Insig1 mRNA.

To examine the effect of HuR on Insig1 mRNA stability, we cloned the Insig1 3′UTR into the downstream of hRluc segment in a psiCHECK2 plasmid (Fig. 6g upper) and then co-transfected this plasmid with HuR or XZ201 control plasmid into 293T cells. 48 h later, we added Actinomycin D to stop the transcription and

harvested a time course to measure the RNA decay rate using Real-time PCR for hRluc. The half-life of hRluc mRNA was increased in the samples overexpressing HuR (HuR $T_{1/2} = 6.7$ h) compared with the control (Control $T_{1/2} = 2.4$ h) (Fig. 6h, full length), while no differences were observed for 18S mRNA and hLuc which was independently transcribed in the plasmid (Fig. 6i, j, full length). Thus, the presence of HuR can enhance the stability of a reporter transcript with Insig1 3′UTR. To test whether the mRNA stabilizing effect is due to the interaction between HuR and mRNA transcript, we removed the two HuR binding sites in the 3′UTR (Fig. 6g lower) and measured its half-life. As expected, the decay rate of the reporter messenger RNAs was not influenced by HuR protein (Fig. 6h–j mutated). These results demonstrate that HuR stabilizes the hRluc mRNA through the interaction with Insig1 3′UTR.

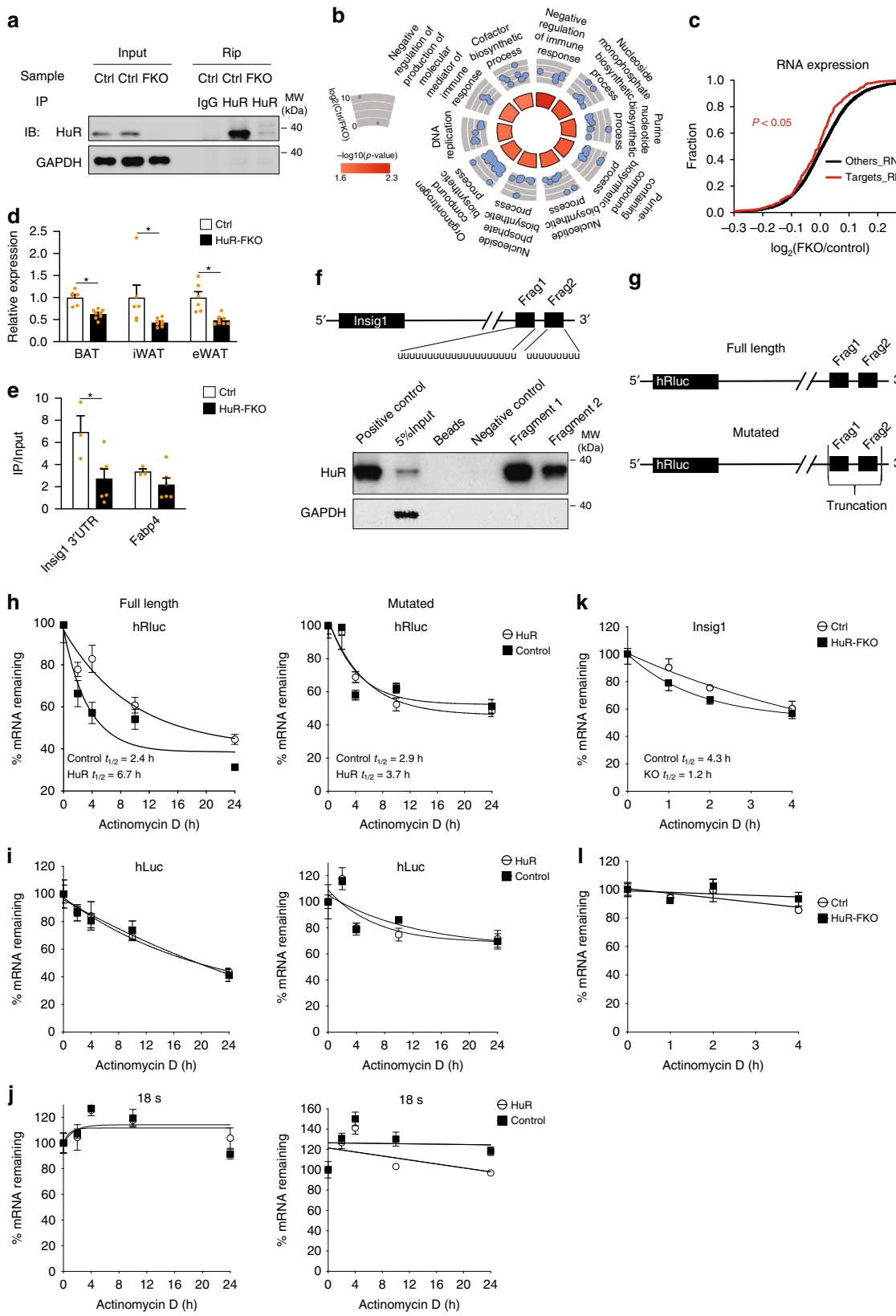

To further investigate the influence of HuR on Insig1 mRNA stability in adipocyte context, we used Actinomycin D to inhibit mRNA transcription in HuR-FKO white adipocyte culture and measured the decay rates for Insig1 mRNA. The successful deletion of HuR was confirmed by genotyping and real-time PCR (Supplementary Fig. 7a, b). The half-life of Insig1 mRNA

decreased from 4.3 to 1.2 h in the absence of HuR (Fig. 6k), while no difference was observed for 18S mRNA (Fig. 6l). Taken together, HuR may repress adipogenesis partially by binding and stabilizing Insig1 mRNA.

To further examined the functional interactions between HuR and Insig1, we overexpressed HuR and knocked down

**Fig. 6 HuR targets and stabilizes Insig1. a** Western blot to confirm the presence of HuR protein in the IP sample by anti-HuR. **b** Circle plot depicting the top 10 most significant biological processes generated using the top 200 most enriched HuR-binding genes in Ctrl. The inner circle is colored by the significance of the GO process. The outercircle represents the scatterplot of the enrichment ($\log_2$(Ctrl/HuR-FKO)) for all the genes found within each GO term. **c** The accumulative fraction curves were plotted for the fold changes of targets and other genes between the FKO and control samples. Kolmogorov–Smirnov test. **d** Real-time PCR to confirm the expression of Insig1 in BAT, iWAT, and eWAT (Ctrl: $n = 6$; HuR-FKO: $n = 8$). **e** RIP assay with anti-HuR in epididymal white adipose tissue lysate from the HuR-FKO and control littermates to examine the amount of Insig1 mRNA in the IP samples. Fabp4 was used as a control. 5% tissue lysate in the IP reaction was used as input (Ctrl, $n = 3$; HuR-FKO, $n = 6$). **f** RNA pull-down assay for the HuR-recognizing RNA segments in Insig1 3′UTR (detailed in Methods). **g** The diagram of a psiCHECK2 reporter with a full length Insig1 3′UTR (upper) or a mutated 3′UTR without the two HuR-binding sites (lower). **h, i** The plasmid diagrammed in (**g**) was co-transfected with HuR overexpression plasmid or control (XZ201) into 293 cells. Actonmycin D was added to stop the transcription, followed by real-time PCR for **h** hRluc, **i** hLuc, and **j** 18 s during a time course. The remaining fractions of hRluc mRNA at each point were fit into a first-phase decay curve to derive the RNA half-life. $n = 6$ per group. **k** Primary white preadipocytes were isolated from HuR-KFO and control animals for culture and then induced to differentiate for 5 days. Actinomycin D was added to track the decay rate. **l** 18S mRNA was used a control. $n = 5$ per group. Error bars are mean ± SEM. Statistical significance was determined by Student's $t$-test; *$p < 0.05$.

Insig1 simultaneously during adipogenesis. Knockdown of Insig1 can indeed attenuate the pro-adipogenic effects from HuR overexpression, indicating that Insig1 is an important downstream target of HuR (Supplementary Fig. 6d).

Beside of insig1, we also examined other potential targets previously reported in other cell types. HuR was reported to promote the commitment of myoblasts to myogenesis by enhancing the translation of HMGB1 and suppressing the translation inhibition mediated by miR-1192[33]. To test whether the enhanced myogenesis pathway in KO BAT is contributed by a decrease of HMGB1, we checked the expression of Hmgb1 in BAT by real-time PCR and western blot but did not find any difference between control and knockout samples (Supplementary Fig. 8a, b). Therefore, the downregulation of muscle markers in FKO BAT is unlikely due to the downregulation of Hmgb1. In addition, Pekala group reported that HuR-targeted C/EBPβ to regulate 3T3-L1 cell differentiation process[24,25]. We conducted real-time PCR and western blot to examine C/EBPβ expression in three adipose tissues, but did not detect any change (Supplementary Fig. 8c, d), highlighting the difference between the in vitro and in vivo systems. Moreover, neither Hmgb1 nor Cebpβ was found in our HuR target candidate list (Supplementary Data 3). Taken together, these findings suggest that the interactions between HuR and its targets are likely to be cell-type specific and environment-dependent.

## Discussion

HuR is an extensively studied RBPs and its function has been determined in a variety of cell types, but its role in metabolism organs still remains unclear. Here, we described a HuR-mediated posttranscriptional regulation in adipogenesis. Knockdown of HuR promotes adipogenesis while overexpression of HuR represses adipogenesis in adipocyte cell culture. Knockdown of HuR in adipose tissue significantly increases fat mass, accompanied by glucose intolerance and insulin resistance. Mechanistically, HuR is able to repress adipogenesis by binding and stabilizing the mRNA of Insig1, a negative regulator in adipogenesis. Taken together, our work identifies HuR as an important posttranscriptional regulator of adipogenesis and provides insights into how posttranscriptional processes contribute to adipocyte development.

Adipogenesis can be largely divided into two phases. First, mesenchymal stem cells commit into adipocyte precursors; second, the adipocyte precursors undergo terminal differentiation to form mature adipocytes filled up with lipid droplets. Because AdipoQ-Cre is not expressed at the precursor stage, our knockout data indicated that HuR should influence the terminal differentiation phase. Moreover, the expression level of HuR is induced at the initial stage during in vitro adipogenesis, but declined in

the later stage to facilitate lipid accumulation (Fig. 1b). Together, our results have demonstrated a repressive role of HuR in the terminal differentiation. However, it still remains unclear whether HuR may play a role in the earlier preadipocyte commitment stage.

Based on a large number of earlier studies, HuR normally exerts its function by targeting and stabilizing ARE-containing transcripts[14–16], and in very rare cases, it may lead to destabilizing effects on its targets[34]. In our study, we indeed observed a significant decrease of HuR's mRNA targets in the knockout adipose tissue (Fig. 6c), consistent with a transcript-stabilizing effect of HuR instead of a destabilizing function. This may seem to be contractionary to a large number of upregulated genes upon HuR knockout (Supplementary Data 3). However, the increased expression of these genes is unlikely due to the destabilization of a series of common transcriptional repressors and mRNA-instability factors in the KO tissue, because HuR, as many other RBPs, should be able to recognize and influence many other targets bearing targeting sites to exert its biological function. To explain the mechanism of a RBP, it is needed to appreciate the complex nature of RBP-targeting mechanism and not to simplify the mechanism using one or two key targets. Given the multiple biological pathways altered in HuR knockout adipose tissues, by no means, we are implying that Insig1 is the sole target and stabilizing transcript is the only effect of HuR on its targets. The phenotype of HuR deficiency in adipose tissues should probably be accounted for by multiple targets. The influence of HuR on its targets may go beyond mRNA stabilization to affect other RNA processing steps such as splicing[14,16]. Therefore, identifying HuR's other targets and understanding how it may affect these targets will warrant further investigation.

As HuR has an RNA-stabilizing effect and its protein level significantly decreases during adipocyte maturation, we investigated whether the HuR protein downregulation may be due to a change in its messenger stability. If this hypothesis is true, the HuR mRNA should be more stable at early than late stage during the differentiation because the HuR protein is more abundant at the early stage (Fig. 1b). We conducted the Actinomycin D experiments to examine the stability of HuR transcript at day 4 and day 8, but we did not detect any difference in the decay rate of HuR mRNA (Supplementary Fig. 9a), indicating an unlikelihood that HuR controls its own expression in adipocytes by stabilizing its own transcripts. To further explore the regulation of HuR expression during adipogenesis, we examined its mRNA expression across adipogenesis by real-time PCR. The mRNA expression of HuR by-and-large matches its protein levels of HuR (Supplementary Fig. 9b). Thus, the abundance change of HuR during adipogenesis should be at least partially due to the transcriptional regulation.

Although our study has demonstrated a negative role of HuR during adipogenesis in all three adipose tissues, it should be noteworthy that the extent of the adipogenesis enhancement varies in three different depots. HuR knockout promotes adipogenesis significantly in white fat depots, which is supported by both real-time PCR and GSEA analysis (Figs. 3 and 4). However, HuR knockout in BAT only results in a mild upregulation of the examined pan-adipogenic markers (Fig. 5b), but did not alter the adipogenesis pathway in the GSEA. Thus, the adipogenic phenotype from HuR deficiency is stronger in WAT than in BAT. In addition, HuR deletion also causes depot-specific phenotypes. For instance, it results in an induced inflammation program in eWAT, an enhanced browning program in iWAT and a repressed myogenesis program in BAT. The depot-specific phenotypes of HuR knockout might be accounted for by at least two mechanisms. First, HuR might target distinct sets of transcripts in different depots. This is consistent with the readily detectable depot-specific transcriptome signature[35,36]. Second, HuR might target similar transcripts in different depots and have similar effects on these targets, but each depot may elicit a depot-specific response program in different depots. These two explanations are non-mutually exclusive. It is possible that HuR targets some depot-specific transcripts and at the same time has a common target set in the different depots that can elicit distinct responses depending on their cellular context. Nonetheless, the mechanism underlying the depot-specific phenotype is of interest for the future study.

## Methods

**Ethics statement**. The human fetal BAT is a generous gift from Zen-Bio Inc. Human fetal BAT was obtained from Advanced Bioscience Resources (Alameda, CA) from deceased donors as approved under exemption 4 in the HHS regulations (45 CFR Part 46). ABR follows established procedures for written informed parental consent. Zen-Bio conducted basic research in accordance with NIH guidelines and the Federal Provisions Pertaining to Research Use of Human Fetal Tissue by NIH Investigators. Zen-Bio's research related to human tissues is approved under its Institutional Review Board (IRB) through PearlIRB. We have complied with all relevant ethical regulations for work with human samples.

**Animal studies**. $HuR^{flox/flox}$ mice were originally purchased from the Jackson Laboratory and subsequently bred in house. The adiponectin-Cre and Ucp1-Cre transgenic mice were gifts from Dr. Evan Rosen at Harvard University. $HuR^{flox/flox}$ mice were backcrossed with C57BL/6 mice for two generations and then used to cross adiponectin-Cre mice to generate AdipoQ-Cre;$HuR^{f/+}$ mice. Heterozygous AdipoQ-Cre;$HuR^{f/+}$ were then crossed with $HuR^{f/f}$ to produce homozygous AdipoQ-Cre;$HuR^{f/f}$. AdipoQ-Cre;$HuR^{f/f}$ were further bred with $HuR^{f/f}$ mice to generate AdipoQ-Cre;$HuR^{f/f}$ (HuR-FKO) and littermate $HuR^{f/f}$ controls (Ctrl) for experiments. The strategy of Ucp1-Cre;$HuR^{f/f}$ (HuR-BATKO) mice generation is similar as AdipoQ-Cre;$HuR^{f/f}$. All mice were kept under specific pathogen-free conditions in the animal vivarium at Duke-NUS Medical School. All animal experimental protocols were approved by the Singapore SingHealth Research Facilities Institutional Animal Care and Use Committee. All relevant ethical guidelines for animal testing and research were followed.

EchoMRI was used to measure fat and lean mass. For glucose tolerance tests (GTT), mice were fasted overnight, followed by intraperitoneal glucose injection (2 g/kg). For the insulin tolerance test (ITT), human insulin (Sigma-Aldrich) was injected (1 unit/kg) to randomly fed mice.

For the in vivo insulin signaling study, HuR-FKO mice were fasted for 6 h at room temperature. The mice were injected with insulin (1 U per kg body weight) and sacrificed after 5 min. Epididymal white adipose tissue was collected to check P-AKT and AKT levels.

Tissues were fixed in 10% formalin and embedded in paraffin. Slides were stained with H&E or immunostained with F4/80 antibody (Bio-Rad, MCA497).

**Human and mouse preadipocyte culture**. Human primary interscapular brown preadipocytes and subcutaneous white preadipocytes (SP-F-1) were obtained from Zen-Bio Inc. All the human cells were cultured and differentiated as described in our earlier studies[26,37] with minor modifications. Briefly, preadipocytes were plated in DMEM supplemented with FBS 10%, penicillin and streptomycin (1:100), and amphotericin B (1 μg/ml). Once cells reached confluence, they are induced to differentiate using differentiation medium (DMEM, FBS 10%, IMBX 0.25 mM, dexamethasone 0.5 mM, insulin 850 nM, indomethacin 100 μM, rosiglitazone 1 μM, penicillin and streptomycin (1:100), and

fungizone/amphotericin B (1 μg/ml). Cells were incubated in differentiation medium for 7 days after which the medium was changed to maintenance medium (DMEM, FBS 10%, insulin 160 nM, penicillin and streptomycin (1:100), and fungizone/amphotericin B (1 μg/ml). Cells were maintained in this manner with medium changed every 2 days.

Primary mouse brown and white preadipocytes were isolated from 3- to 4-week-old C57BL6 mice. Preadipocytes isolation, culture and differentiation were conducted as previous studies with minor modifications[13,36,38]. Briefly, interscapular BATs or iWATs from ~3-week-old mice were minced, and digested in 0.2% collagenase, which were subsequently filtered by 40 μm cell strainer and centrifuged to collect stromal vascular fraction (SVF) cells in the pellets. SVF cells were cultured for downstream experiments. Primary SVF cells were cultured in DMEM with 5% new-born calf serum and 5% FBS until confluence. Cells were induced to differentiate for 2 days with DMEM containing 10% FBS, 850 nM insulin (Sigma), 0.5 μM dexamethasone (Sigma), 250 μM 3-isobutyl-1-methylxanthine, phosphodiesterase inhibitor (IBMX, Sigma), and 1 μM rosiglitazone (Cayman Chemical). The induction medium was replaced with DMEM containing 10% FBS and 160 nM insulin for 2 days. Then cells were incubated in DMEM with 10% FBS.

**Retrovirus transduction**. 293T cells (ATCC) for retroviral packing were cultured in DMEM containing 10% FBS (HyClone). The retroviral vector pSUPER (Oligoengine) was used to generate short hairpin (sh)RNAs to infect preadipocytes; XZ201 vector[13] was used to overexpress HuR for gain-of-function studies. All of the retroviruses were packaged in 293T cells with the pCL-eco (IMGENEX/RetroMax) packaging vector for mouse preadipocyte infection and with pCL-10A (IMGENEX/RetroMax) packaging vector for human preadipocyte infection. X-tremeGENE9 Transfection Reagent (Roche) was used for plasmid transfection according to the manufacturer's instructions. Retroviruses were used to transduce preadipocytes in the presence of 4 mg/mL Polybrene (Sigma-Aldrich), followed by induction of differentiation.

**Quantitative real-time PCR and western blot**. Total RNA from mouse tissues was extracted with miRNeasy Mini Kit (Qiagen). RT-PCR was performed using M-MLV Reverse Transcriptase kit (Promega) with random primers. Real-time PCR was performed on ViiA$^{Tm}$7 system (Applied Biosystems) using SYBR Green PCR Master Mix (Bioline). Gene expression was quantified by the comparative threshold cycle (ΔΔCT) method and results were normalized to the expression of Rpl23 RNA. The primers for Real-time PCR are listed in Supplementary Data 1. Western blotting was performed to detect target proteins using HuR (1:1000, sc-5261, Santa Cruz Biotechnology), GAPDH (1:5000, ab8245, Abcam), UCP1 (1:2000, ab10983, Abcam), PGC1α (1:1000, sc-13067, Santa Cruz Biotechnology), DESMIN (1:1000, MA1-06401, Thermo Fisher Scientific), CIDEA (1:1000, sc-366814, Santa Cruz Biotechnology), phosphorylated (p)-AKT (1:1000, 4058S, Cell Signaling), AKT (1:2000, 9272S, Cell Signaling), INSIG1 (1:1000, sc-98984, Santa Cruz), C/EBPbeta (1:2000, ab32358, Abcam), HMGB1 (1:1000, ab79823, Abcam), and Beta-Actin (1:5000, A1978, Sigma-Aldrich) antibodies.

**RNA immunoprecipitation**. Tissue lysates were prepared from eWAT of Ctrl and HuR-FKO in lysis buffer (Tris, pH 7.4 20 mM, MgCl$_2$ 5 mM, NaCl 150 mM, GA-630 1%, DTT 1 mM, heparin 0.50 mg/ml, SUPERaseIn 0.2 unit/μl). The subsequential RNA immunoprecipitation (RIP) was performed using the Magna RIP kit (Merck Millipore) according to the manufacturer's instruction. RNA samples retrieved from anti-HuR were used for RNA-seq.

**RNA pull-down**. RNA pull-down was performed according to our published protocol with minor modifications[13,39]. The mouse eWAT lysate was prepared as described in RNA Immunoprecipitation above. Segments in 3′UTR as shown in the diagram (Fig. 6f) were cloned for in vitro transcription to generate RNA fragments which were used for RNA pull-down assay in eWAT lysate. An AU-enriched ~100 nt fragment from androgen receptor was used as a positive control. Biotin-labeled RNA fragments were generated using a MEGAscript kit (Life Technologies) in the presence of biotin-CTP (Thermo Fisher Scientific) and purified with Nuc-Away spin columns. The biotin-labeled fragments were conjugated with Dynabeads and incubated in tissue lysate for 3 h at 4 °C. Dynabeads were washed with washing buffer three times (Tris 20 mM, pH 7.4, MgCl$_2$ 5 mM, NaCl 150 mM, GA-630 0.05%) and boiled in western blot loading buffer for 5 min at 95 °C. The retrieved proteins were visualized by immunoblotting.

**Ribosome profiling and data analysis**. For ribosome profiling, eWAT from control and KO mice were homogenized with pestle and mortar in the presence of liquid nitrogen. The homogenized powder was transferred into 1.5 ml tubes and lysed with 1 ml lysis buffer (1% NP-40, 200 mM KOAc, 25 mM K-HEPES pH 7.2, 14 mM MgCl, and 4 mM CaCl$_2$). Samples were incubated on ice for 5 min and centrifuged at 9000 × $g$ in a benchtop centrifuge at 4 °C for 10 min. The fat layer on top was removed and the 300 μl supernatant was collected. Following steps were conducted according to a published protocol[40].

**RNA decay analysis**. 293 cells or adipocyte cultures were treated with 5 mg/mL actinomycin D (Sigma-Aldrich) and harvested RNA at different times as indicated in the figures. The same proportion of RNA was taken at different time points for real-time PCR. CTs from each sample were used to calculate the remaining percentage of mRNA at each point. These data are fit into a first-phase decay model to derive mRNAs' half-life, $Y_t = (Y_0 - \text{Plateau})*\exp(-k_{\text{decay}}*t) + \text{Plateau}$ with $Y_t$ representing the remaining percentage at a given time, $Y_0$ representing the initial amount of RNA, $t$ representing time after transcription inhibition, and $k_{\text{decay}}$ indicating the rate constant.

**RNA-seq analysis**. 3 HuR-FKO and 3 Control samples were used for eWAT and iWAT RNA-seq. 2 HuR-FKO and 2 Control samples were used in BAT RNA seq. The RNA of each sample was pooled from three individual animals. Strand-specific RNA-seq libraries were prepared using NEBNext Ultra RNA Library Prep Kit for Illumina and sequenced on the Illumina Hiseq4000 platform. The paired-end reads for BAT, eWAT, and iWAT were aligned against mm10 using Tophat v2.0.11[41]. The aligned files were passed into Cufflinks-2.1.1 for gene quantification into FPKM units. For each tissue, the average FPKM values were evaluated for both HuR-FKO and Ctrl groups. Protein-coding genes with low expression (Average FPKM ≤ 1) in both groups were removed from all downstream analyses.

**Gene set enrichment analysis**. Protein-coding genes were first pre-ranked by log2 (HuR-FKO/Ctrl) expression changes for each tissue and loaded into GSEA to identify enriched biological process by comparing against "Hallmarks" gene sets in MSigDB using default parameters. All the sequencing data generated in this study have been deposited in GEO database (GSE124280).

**Statistical analysis**. Data are presented as mean ± SEM. Statistical significance was assessed using the unpaired, two-tailed Student $t$ test. Statistical significance for samples with more than two groups was determined by one-way ANOVA. The distribution difference between different cumulative curves was determined by Kolmogorov–Smirnov test. $P$ values of <0.05 were considered to be significant.

## Data availability

All generated sequencing data have been deposited into GEO database with accession number GSE124280. All data are available from the corresponding author upon reasonable request and that a Source Data file is available for this article.

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

## Acknowledgements

This work was supported by Singapore National Medical Research Council's Open Fund - Young Individual Research Grant (OFYIRG) (NMRC/OFYIRG/0080/2018),

Open Fund-Individual Research (OF-IRG) Grant (NMRC/OFIRG/0062/2017), Ministry of Education (MOE) Tier2 grant (MOE2017-T2-2-015), National Medical Research Council's Cooperative Basic Research Grant (CBRG; NMRC/CBRG/0070/2014 and NMRC/CBRG/0101/2016) and Tanoto Initiative in Diabetes Research to L.S.

## Author contributions

D.X., D.T.C.S.,Y.C.L., A.M.M.K., K.N.W., S.Y.C., U.D., B.C.T. and A.C.E.W. performed experiments. D.X. and L.S. designed experiments and wrote the paper. X.H. discussed the experiment design and critically reviewed the paper. L.S. is the guarantor of this work and, as such, had full access to all the data in the study and takes responsibility for the integrity of the data and the accuracy of the data analysis.

## Competing interests

The authors declare no competing interests.
