## [Peer Review File · Nature Communications]

Reviewers' comments:

Reviewer #1 (Remarks to the Author):

In their manuscript entitled "The RNA-binding protein HuR is a negative regulator in adipogenesis" Sing et al. elucidated the functional role of the ubiquitous RNA-binding protein HuR in adipogenesis in primary adipocyte culture as well as by fat-specific HuR knockout mice. The authors convincingly demonstrate that knockdown of HuR promotes brown fat marker and pan-adipocyte marker expression and is accompanied by an increase in lipid accumulation in differentiating preadipocytes implicating that HuR represents a key repressor for both brown adipose tissue (BAT) and white adipose tissue (WAT). Consistently, HuR levels are strongly decreased in mature adipocyte of three different adipose tissues (brown fat, inguinal and epididymal white fat). Furthermore, adipose-selective HuR ablation results in increased fat mass, glucose intolerance and insulin resistance thus supporting the critical role of HuR in the control of key metabolic functions. In addition, fat specific knockout of HuR caused an increase in body fat mass mainly due to an enlarged epididymal white depot also implicated by the hypertrophy in white fat tissue and an impaired insulin sensitivity (suggested by a reduction in AKT phosphorylation). Profiling of changes in global gene expression patterns in iWAT from HuR knockout mice revealed a strong upregulation of adipogenesis genes, fatty acid metabolism pathways and oxidative phosphorylation, respectively. Importantly, results from ribosome profiling indicate that HuR acts mainly on the mRNA level of target genes with only little effects on translation. By using RNA pull down assay, the authors present insulin induced gene 1 (Insig1) as a major target of HuR dependent mRNA-stabilization. Insig1 is a negative regulator of adipogenesis. From these data the authors speculate that HuR represses adipogenesis partially through Insig 1. Finally, the authors demonstrate that overexpression of HuR in adipose tissue protects from adipose tissue expansion under high fat diet implicating that HuR may protect against obesity and metabolic syndromes.

This is a very comprehensive and in depth study with many interesting findings and data supporting the overall strong relevance of the RNA-binding protein on gene expression profiles of adipose tissues and adipocyte differentiation and maturation. Regardless of the overall high quality and set-up of this extensive study, there is one critical aspect concerning the mechanisms of HuR actions which is not adequately addressed.

Specifically,

1. The major conclusions of this study are based on data from differential expression profiles of adipocyte marker expression (Fig. 1) and GSEA from eWAT (Fig. 3) and iWAT (Fig. 4) upon genetic HuR depletion. The data show a clear increase in the expression of many "hallmark" gene sets upon HuR depletion which implicates a suppressive effect of HuR on these genes. Data from ribosome profiling revealed that HuR has obviously no major impact on the translation of target mRNAs supporting that the major effects of HuR occur predominantly on the mRNA level. The main question which arises and which is not addressed in the current version are the possible mechanisms underlying HuR suppression of all of these target mRNAs. I do absolutely agree with the authors that Insig1 is not the sole reason for all of these effects – Importantly, HuR exclusively exerts stabilizing effects on ARE-containing mRNAs and only in very rare cases, indirectly and via recruitment of some specific miRNAs can induce an increase in mRNA decay. Therefore, the observed induction of this high number of target genes upon HuR knockdown to my opinion can only be explained by different scenarios. HuR may either stabilize a common transcriptional repressor of these genes or, alternatively, it may stabilize an mRNA-instability factor controlling all of these mRNA targets. Both scenarios seem rare. To shed more light on this issue, the authors should focus on this aspect in the discussion but in addition, the following questions should be addressed by further experiments and analysis:

- What is the molecular basis for the downregulation of HuR during adipocyte maturation? transcriptional effects or rather effects on mRNA-stability? since HuR is able to stabilize its own message, I am wondering whether HuR mRNA-stability is decreased during adipocyte differentiation. Act-D experiments can answer this question.
- P6, para 1. The authors can simply check their hypothesis whether knockdown of HuR results in an increased migration of immune cells within the larger intracellular spaces by IF staining of F4/80.
- Do some of the direct HuR target mRNAs identified in eWAT by RIP assay (Fig. 6E) also

accumulate in the gene set enrichment analysis (GSEA) shown in Figure 3 and are those mainly increased upon HuR knockdown ?

- What is the number of animals used for assessment of p values in the enrichment scores shown in Fig. 3E,F, 4C,D,H and Fig. 5A ?

- Fig. 5C. P7, second para. The authors state that "the down-regulation of muscle markers is mainly a consequence of an enhanced commitment to the brown fat lineages since both share the same precursors. I assume that downregulation of HuR is a direct consequence of reduced myogenesis by the HMGB1 protein which was identified as a direct target of HuR-dependent mRNA stabilization (Dormoy-Raclet et al., Nat. Commun. 2013). In Fig. 5C, authors should therefore check for HMGB1 contents.

- Fig. 5D, in addition to showing upregulation of some BAT markers, the authors should show reduction of some myogenesis markers on the protein level.

- P. 7, first para. What are the underlying mechanisms of the depot-specific difference in gene expression profiles upon HuR knockdown although although genes of fatty acid metabolism and adipogenesis are similarly affected? Depot-specific posttranslational HuR modifications which may critically determine whether HuR can bind a specific target or not may play a crucial role. This issue needs to be discussed.

- Fig. 6B. Please give a list of top candidates of HuR-bound genes of each segment. Which of these bear typical ARE signatures within their 3'UTR? (the analysis by "UTR Scan" program may help)

- In addition to C/EBP α , the authors should monitor the levels of C/EBP β as a marker for adipogenesis. Importantly, C/EBP β is a target of HuR-mediated mRNA stabilization (Gantt K et al., 2005).

- Considering the fact that the study is already very comprehensive and to my opinion, clearly overloaded for a "Communication", data summarized in Fig. 7 should not be included since they address another aspect which is not that relevant for the story. An overexpression of HuR does not represent physiological conditions of adipose tissues, the opposite is the case (see Fig. 1A).

- Fig. 7G-J. Please, show levels of Insig 1 mRNA

- The Discussion is too concise. The authors should discuss the possible mechanisms how HuR negatively affects adipocyte gene profiles.

Minor:

1. Please use consistently either "HuR" or "HUR"

2. P 10, last para: ..." we by no mean to imply that..." please change into "by no means, we are implying that Insig 1..."

3. P6, para 2. Please explain the meaning of "beige adipocytes" for those readers who are not familiar with this topic?

4. The authors should cite and discuss an earlier review from Cherry et al., 2006 as it describes a opposite role of HuR protein which is necessary for the differentiation of 3T3-L1 preadipocytes.

Reviewer #2 (Remarks to the Author):

This study by Siang and colleagues investigated an RNA-binding protein HuR in adipocyte and have convincingly demonstrated its function in repressing adipogenesis by using primary cell culture models, two adipocyte conditional KO models, one adipocyte transgenic model, plus comprehensive transcriptome analyses and RNA regulation studies. This study is important and significant from these three aspects: 1. HuR is an important RNA binding protein and now its function is extended to regulating adipocyte biology; 2. RNA-related regulation is emerging as an important aspect in adipocyte field and the current study may stimulate this trend by establishing a repressive role of HuR in adipogenesis and fat remodeling; 3. One finding in this study is very interesting and important, that is the bifurcation of insulin sensitivity and browning. HuR-FKO is actually more insulin resistant and glucose intolerant but with more browning. This bifurcation raises alarm to the current mainstream that browning is always associated with improved insulin sensitivity and metabolism, and is noticed by more and more groups but without paying enough attention. From this perspective, this study should be appreciated as an "honest" and objective

study. In addition, the data is presented at high quality, complete and compelling.

Comments:

1. The data in Fig.1 is clean, complete, compelling, and even includes human primary brown and white adipocyte models to further strengthen the conclusion.

2. The more pronounced enlargement in visceral fat of HuR-FKO mice is interesting and makes sense. It is probably due to a balance between adipogenesis and browning. KO of HuR increased both, but the brown remodeling in BAT and iWAT is more active, and thus compensates the hypertrophy owing to increased adipogenesis. To elaborate this point, the brown markers and lipid oxidative pathway should be included in RNA-seq of eWAT in Fig. 3. The authors briefly compared in Fig. 4H but need more details such as gene expression validation by RT-PCR.

3. The BAT KO by Ucp1-Cre is a very interesting model. The KO have promoted browning and adipogenesis in BAT and mildly in iWAT, corresponding to the efficiency of Ucp1-Cre. What happens to eWAT? Possibly no phenotype because of no deletion. Then do the mice develop insulin resistance and glucose intolerance as the KO by Adipoq-Cre? This is quite informative to dissect the depot contribution to overall metabolic phenotype, and further support the bifurcation of browning and insulin sensitivity.

4. The mechanistic studies are solid and insightful. The authors performed RIP-seq to identify targets, then ribosome profiling to exclude the possible effects on translation. Their study on the regulation of Insig-1 by HuR is a strength in this manuscript. It is clearly demonstrated on molecular levels by multiple complementary approaches. However, it is not complete. The author should perform rescue experiment to assure the contribution by Insig-1, such as overexpressing Insig-1 without the nascent 3-UTR in HuR KO, or knockdown Insig-1 in HuR overexpression cells.

5. The findings in the overexpression model of HuR by aP2 promoter are complementary and interesting. There is no effect in chow-fed mice when adipogenesis is at the basal level but KO showed reduced bodyweight gain on HFD feeding when adipogenesis is stimulated. What about inflammation in eWAT? It is expected to decrease according to the KO mice data in Fig. 3. But, if the reduced adipogenesis causes a defect in handling and storing excessive lipid on HFD feeding, the effects may be neutralized. It will be interesting to examine inflammation in eWAT.

6. The body composition in Fig. 7B should be presented in absolute weight rather than percentage since their bodyweight is different.

7. In Fig. 7F: Error bars are missing? The mice are still quite insulin sensitive, meaning the HFD-induced insulin resistance has not been fully established, thus only mild effects on improving insulin sensitivity in the transgenic mice. Same to the insignificant improvement on GTT in Fig. S7J.

8. The authors should pay more attention to language. Three typos are found in the Abstract: an RNA; inguinal white, and brown adipose tissue; (a) new insight. There are more in the manuscript.

9. Missing data in Fig 2G.

Reviewers' comments:

Reviewer #1 (Remarks to the Author):

In their manuscript entitled "The RNA-binding protein HuR is a negative regulator in adipogenesis" Sing et al. elucidated the functional role of the ubiquitous RNA-binding protein HuR in adipogenesis in primary adipocyte culture as well as by fat-specific HuR knockout mice. The authors convincingly demonstrate that knockdown of HuR promotes brown fat marker and pan-adipocyte marker expression and is accompanied by an increase in lipid accumulation in differentiating preadipocytes implicating that HuR represents a key repressor for both brown adipose tissue (BAT) and white adipose tissue (WAT). Consistently, HuR levels are strongly decreased in mature adipocyte of three different adipose tissues (brown fat, inguinal and epididymal white fat). Furthermore, adipose-selective HuR ablation results in increased fat mass, glucose intolerance and insulin resistance thus supporting the critical role of HuR in the control of key metabolic functions. In addition, fat specific knockout of HuR caused an increase in body fat mass mainly due to an enlarged epididymal white depot also implicated by the hypertrophy in white fat tissue and an impaired insulin sensitivity (suggested by a reduction in AKT phosphorylation). Profiling of changes in global gene expression patterns in iWAT from HuR knockout mice revealed a strong upregulation of adipogenesis genes, fatty acid metabolism pathways and oxidative phosphorylation, respectively. Importantly, results from ribosome profiling indicate that HuR acts mainly on the mRNA level of target genes with only little effects on translation. By using RNA pull down assay, the authors present insulin induced gene 1 (Insig1) as a major target of HuR dependent mRNA-stabilization. Insig1 is a negative regulator of adipogenesis. From these data the authors speculate that HuR represses adipogenesis **partially** through Insig 1. Finally, the authors demonstrate that overexpression of HuR in adipose tissue protects from adipose tissue expansion under high fat diet implicating that HuR may protect against obesity and metabolic syndromes.

This is a very comprehensive and in depth study with many interesting findings and data supporting the overall strong relevance of the RNA-binding protein on gene expression profiles of adipose tissues and adipocyte differentiation and maturation. Regardless of the overall high quality and set-up of this extensive study, there is one critical aspect concerning the mechanisms of HuR actions which is not adequately addressed.

Specifically,

1. The major conclusions of this study are based on data from differential expression profiles of adipocyte marker expression (Fig. 1) and GSEA from eWAT (Fig. 3) and iWAT (Fig. 4) upon genetic HuR depletion. The data show a clear increase in the expression of many "hallmark" gene sets upon HuR depletion which implicates a suppressive effect of HuR on these genes. Data from ribosome profiling revealed that HuR has obviously no major impact on the translation of target mRNAs supporting that the major effects of HuR occur predominantly on the mRNA level. The main question which arises and which is not addressed in the current version are the possible mechanisms underlying HuR suppression of all of these target mRNAs. I do absolutely agree with the authors that Insig1 is not the sole reason for all of these effects – Importantly, HuR exclusively exerts stabilizing effects on ARE-containing mRNAs and only in very rare cases, indirectly and via recruitment of some specific miRs, can induce an increase in mRNA decay. Therefore, the observed induction of this high number of target genes upon HuR knockdown to my opinion can only be explained by different scenarios. HuR may either stabilize a common transcriptional repressor of these genes or, alternatively, it may stabilize an mRNA-instability factor controlling all of these mRNA targets. Both scenarios seem rare. To shed more light on this issue, the authors should focus on this aspect in the discussion but in addition, the following questions should be addressed by further experiments and analysis:

Thanks for the insightful comments from the reviewer. We agree with the reviewer that the mechanism of HuR in adipocytes is likely to stabilize its target and the target recognition is complicated. According to the reviewer's suggestion, we have further discussed the HuR-mRNA interaction mechanism and the potential consequential effect on its targets.

"Based on a large number of earlier studies, HuR normally exerts its function by targeting and stabilizing ARE-containing transcripts^{14, 15, 16}, and, in very rare cases, it may lead to destabilizing effects on its targets³⁴. In our study, we indeed observed a significant decrease of HuR's mRNA targets in the knockout adipose tissue (Figure 6C), consistent with a transcript-stabilizing effect of HuR instead of a destabilizing function. This may seem to be contractionary to the large number of up-regulated genes upon HuR knockout (supplemental file 3). However, the increased expression of these genes is unlikely due to the destabilization of a series of common transcriptional repressors and mRNA-instability factors in the KO tissue, because HuR, as many other RBPs, should be able to recognize and influence many other targets bearing targeting sites to exert its biological function. To explain the mechanism of a RBP, it is needed to appreciate the complexity nature of RBP-targeting mechanism and not to simplify the mechanism using one or two key targets. Given the multiple biological pathways altered in HuR knockout adipose tissues, by no means, we are implying that Insig1 is the sole target and stabilizing transcript is the only effect of HuR on its targets. The phenotype of HuR deficiency in adipose tissues should probably be accounted by multiple targets. The influence of HuR on its targets may go beyond mRNA stabilization to affect other RNA processing steps such as splicing^{14, 16}. Therefore, identifying HuR's other targets and understanding how it may affect these targets will warrant further investigation."

- What is the molecular basis for the downregulation of HuR during adipocyte maturation? transcriptional effects or rather effects on mRNA-stability? since HuR is able to stabilize its own message, I am wondering whether HuR mRNA-stability is decreased during adipocyte differentiation. **Act-D experiments can answer this question.**
Thanks for this suggestion. We agree that the regulation of HuR protein during adipogenesis is worth further investigation.

"As HuR has a RNA-stabilizing effect and its protein level significantly decreases during adipocyte maturation, we investigated whether the HuR protein downregulation may be due to a change in its messenger stability. If this hypothesis is true, the HuR mRNA should be more stable at early than late stage during the differentiation because the HuR protein is more abundant at the early stage (Figure 1B). We conducted the Actinomycin D experiments to examine the stability of HuR transcript at day 4 and day 8, but we did not detect any difference in the decay rate of HuR mRNA (Figure S9A), indicating an unlikelihood that HuR controls its own expression in adipocytes by stabilizing its own transcripts. To further explore the regulation of HuR expression during adipogenesis, we examined its mRNA expression across adipogenesis by Real-time PCR. The mRNA expression of HuR by-and-large matches its protein levels of HuR (Figure S9B). Thus, the abundance change of HuR during adipogenesis should be at least partially due to the transcriptional regulation."

We have included these data in our revised manuscript. (Figure S9).

- P6, para 1. The authors can simply check their hypothesis whether knockdown of HuR results in an increased migration of immune cells within the larger intracellular spaces by IF staining of F4/80.

It is a good point. We have conducted the staining experiment. The IF staining of F4/80 clearly indicates that more macrophages are presented in the KO eWAT. We have included this data in the revised manuscript (Figure 2F).

- Do some of the direct HuR target mRNAs identified in eWAT by RIP assay (Fig. 6E) also accumulate in the gene set enrichment analysis (GSEA) shown in Figure 3 and are those mainly increased upon HuR knockdown ?

As suggested, we have independently intersected HuR targets with downloaded MSigDB reference gene sets of (i) Hallmark_Interferon Gamma Response, (ii) Hallmark_Adipogenesis and (iii) Hallmark_Inflammatory Response. However, we noticed only a low overlap of three (Bst2, Pnp, Traf1d1), two (Aco2, Dnajb9) and three genes (Bst2, Myc, Rhog) with the above named enriched GSEA respectively (Figure S4A-C). To ensure the low overlapping rate is not a result of the arbitrary cut-off imposed to define HuR targets (top 200 most enriched by HuR immunoprecipitation), we compared the overall HuR enrichment distributions between reference gene set and its corresponding complementary HuR-bound transcript set. Across the three investigated enriched GSEA processes shown from Figure 3, we found 58, 163 and 47 genes to exhibit detectable levels of HuR binding (Figure S4D-F). For each GSEA process, we compared the HuR enrichment profiles between the above identified genes and the complementary set (out of all HuR-bound transcripts). However, none of the comparisons yielded any significant differences (Figure S4G-I).

We have included these data in the revised manuscript (Figure S4)

“HuR targets identified in our RIP-seq analysis showed little overlapping with the genes in the up-regulated GSEA pathways in **Figure 3** (Figure S4 A-I), which suggests that the up-regulated pathways in **Figure 3** are likely due to a downstream effect from HuR’s direct targets.”

Figure S4 Overlaps of the HuR targets with the genes in the GSEA.

(A-C) Overlaps of HuR targets with downloaded MSigDB reference gene sets of (A) Hallmark_Interferon Gamma Response, (B) Hallmark_Adipogenesis and (C) Hallmark_Inflammatory Response. (C-D) Proportion of genes within reference gene sets of (D) Hallmark_Interferon Gamma Response, (E) Hallmark_Adipogenesis and (F) Hallmark_Inflammatory Response with detectable HuR binding. (G-I) Cumulative density functions of the genes with detectable HuR binding identified from (D-F) for (G) Hallmark_Interferon Gamma Response, (H) Hallmark_Adipogenesis and (I) Hallmark_Inflammatory Response.

- What is the number of animals used for assessment of p values in the enrichment scores shown in Fig. 3E,F, 4C,D,H and Fig. 5A ?

We have included the animal numbers in Methods. "3 HuR-FKO and 3 Control samples were used for eWAT and iWAT RNA seq. 2 HuR-FKO and 2 Control samples were used in BAT RNA seq. The RNA of each sample was pooled from 3 individual animals."

We employed the "preranked" function in our GSEA analysis. The P value is for the gene expression shift in the pathways and is independent from the animal number. We did further experimentally validate the key markers in the significant pathways detected in GSEA (Figure 3C, D, G; Figure 4 E, F, G; Figure 5B, C, D, E, G).

- Fig. 5C. P7, second para. The authors state that "the down-regulation of muscle markers is mainly a consequence of an enhanced commitment to the brown fat lineages since both share the same precursors. I assume that downregulation of HuR is a direct consequence of reduced myogenesis by the protein which was identified as a direct target of HuR-dependent mRNA stabilization (Dormoy-Raclet et al., Nat. Commun. 2013). In Fig. 5C, authors should therefore check for HMGB1 contents.

Thanks for this suggestion. Accordingly, we checked the expression of Hmgb1 by Q-PCR, but didn't find any difference between control and knockout samples. Therefore, the down-regulation of muscle markers is unlikely due to the down-regulation of HMGB1 but likely due to an enhanced brown adipogenesis. We have included this data in our revised manuscript (Figure S8A) to emphasize the point that the interactions between HuR and its targets are cell type-specific and environment-dependent.

- Fig. 5D, in addition to showing upregulation of some BAT markers, the authors should show reduction of some myogenesis markers on the protein level.

We have conducted a Western blot for the Des gene and shown a reduction at the protein level. We have included this data in the revised manuscript (Figure 5D).

- P. 7, first para. What are the underlying mechanisms of the depot-specific difference in gene expression profiles upon HuR knockdown although although genes of fatty acid metabolism and adipogenesis are similarly affected? Depot-specific **posttranslational HuR modifications** which may critically determine whether HuR can bind a specific target or not may play a crucial role. This issue needs to be discussed.

We agree with the reviewer that the depot-specific posttranscriptional HuR modification should be further discussed. We have discussed this point in detail in the revised manuscript. "Although our study has demonstrated a negative role of HuR during adipogenesis in all three adipose tissues, it should be noteworthy that the extent of the adipogenesis enhancement varies in three different depots. HuR knockout promotes adipogenesis significantly in white fat depots, which is supported by both real-time PCR and GSEA analysis (Figure 3, 4). However, HuR knockout in BAT only results in a mild upregulation of the examined pan-adipogenic markers (Figure 5B), but did not alter the adipogenesis pathway in the GSEA. Thus, the adipogenic phenotype from HuR deficiency is stronger in WAT than in BAT. In addition, HuR deletion also causes depot-specific phenotypes. For instance, it results in an induced inflammation program in eWAT, an enhanced browning program in iWAT and a repressed myogenesis program in BAT. The depot-specific phenotypes of HuR knockout might be accounted by at least two mechanisms. First, HuR might target distinct sets of transcripts in different depots. This is consistent with the readily detectable depot-specific transcriptome signature^{35, 36}. Second, HuR might target similar transcripts in different depots and have similar effects on these targets, but each depot may elicit a depot-specific response program in different depots. These two explanations are non-mutually exclusive. It is possible that HuR targets some depot-specific transcripts and at the same time has a common target set in different depot that can elicit distinct response

depending to their cellular context. Nonetheless, the mechanism underlying the depot-specific phenotype is of interest for the future study.”

- Fig. 6B. Please give a list of top candidates of HuR-bound genes of each segment. Which of these bear typical ARE signatures within their 3'UTR? (the analysis by “UTR Scan” program may help)

To identify which of these genes bore ARE signature, we overlapped them with genes downloaded from AREsite2 database (<http://rna.tbi.univie.ac.at/AREsite2/bulk>), which is a computational and experimental collection of UTRs bearing ARE motifs. Here, we only considered genes with plain ATTTA pentamer. Results have been summarised in the table below. These data are included in the revised manuscript as Supplemental File 5.

Term	Number of genes	#Genes with ARE (ATTTA pentamer) in 3'UTR
Negative regulation of immune response	6	2
Nucleoside monophosphate biosynthetic process	7	6
Purine nucleotide biosynthetic process	9	7
Purine containing compound biosynthetic process	9	7
Nucleotide biosynthetic process	10	8
Nucleoside phosphate biosynthetic process	10	8
Organonitrogen compound biosynthetic process	33	21
DNA replication	8	4
Negative regulation of production of molecular mediator of immune response	3	1
Cofactor biosynthetic process	8	5
Others	148	97

- In addition to C/EBP α , the authors should monitor the levels of C/EBP β as a marker for adipogenesis. Importantly, C/EBP β is a target of HuR-mediated mRNA stabilization (Gantt K et al., 2005).

As suggested, we have conducted real-time PCR to examine C/EBP β , but its expression is not affected, which also highlights the point that the RBP-mRNA interactions may depend on the cellular context. We have included the C/EBP β in the revised manuscript (Figure S8B)

- Considering the fact that the study is already very comprehensive and to my opinion, clearly overloaded for a “Communication”, data summarized in Fig. 7 should not be included since they address another aspect which is not that relevant for the story. An overexpression of HuR does not represent physiological conditions of adipose tissues, the opposite is the case (see Fig. 1A).

- Fig. 7G-J. Please, show levels of Insig 1 mRNA

We absolutely agree with the reviewer that the transgenic mouse data probably doesn't represent a physiological condition and this study has already been overloaded. Thus, we removed all the transgenic data in the revised manuscript.

- The Discussion is too concise. The authors should discuss the possible mechanisms how HuR negatively affects adipocyte gene profiles.

As suggested by the reviewers, we have further developed the discussion section regarding the possible mechanisms of HuR with a focus on the points of multiple target recognition and depot specific effect.

Minor:

1. Please use consistently either “HuR” or “HUR”

We have fixed this.

2. P 10, last para: ..” we by no mean to imply that...” please change into “by no means, we are implying that Insig 1...”

We have fixed this.

3. P6, para 2. Please explain the meaning of “beige adipocytes” for those readers who are not familiar with this topic?

We have fixed this.

“iWAT is often referred to as beige fat because it is enriched for beige adipocytes that are dispersed in iWAT. These beige adipocytes exhibit white adipocytes features at thermoneutrality but can undergo browning phenotype to take on BAT-like cellular and molecular phenotypes under environmental stimuli such as cold exposure”

4. The authors should cite and discuss an earlier review from Cherry et al., 2006 as it describes an opposite role of HuR protein which is necessary for the differentiation of 3T3-L1 preadipocytes.

We have added this reference in the introduction.

Reviewer #2 (Remarks to the Author):

This study by Siang and colleagues investigated an RNA-binding protein HuR in adipocyte and have convincingly demonstrated its function in repressing adipogenesis by using primary cell culture models, two adipocyte conditional KO models, one adipocyte transgenic model, plus comprehensive transcriptome analyses and RNA regulation studies. This study is important and significant from these three aspects: 1. HuR is an important RNA binding protein and now its function is extended to regulating adipocyte biology; 2. RNA-related regulation is emerging as an important aspect in adipocyte field and the current study may stimulate this trend by establishing a repressive role of HuR in adipogenesis and fat remodeling; 3. One finding in this study is very interesting and important, that is the bifurcation of insulin sensitivity and browning. HuR-FKO is actually more insulin resistant and glucose intolerant but with more browning. This bifurcation raises alarm to the current mainstream that browning is always associated with improved insulin sensitivity and metabolism, and is noticed by more and more groups but without paying enough attention. From this perspective, this study should be appreciated as an “honest” and objective study. In addition, the data is presented at high quality, complete and compelling.

Comments:

1. The data in Fig.1 is clean, complete, compelling, and even includes human primary brown and white adipocyte models to further strengthen the conclusion.

Thanks for the reviewer’s positive comment.

2. The more pronounced enlargement in visceral fat of HuR-FKO mice is interesting and makes sense. It is probably due to a balance between adipogenesis and browning. KO of HuR increased both, but the brown remodeling in BAT and iWAT is more active, and thus compensates the hypertrophy owing to increased adipogenesis. To elaborate this point, the brown markers and lipid oxidative pathway should be included in RNA-seq of eWAT in Fig. 3. Thanks for this suggestion. The pathways showed in this figures are the significantly ones in the GSEA otherwise the figure will be too crowded. Unfortunately, lipid oxidative pathway was not significantly enriched when we performed GSEA and hence not present in this figure. Actually, the eWAT didn’t exhibit a browning phenotype in the GSEA analysis.

The authors briefly compared in Fig. 4H but need more details such as gene expression validation by RT-PCR.

The adipogenesis markers have been examined by Real-time PCR in the original manuscript. As suggested by the reviewer, we also examined the possible change of BAT markers in eWAT. We have provided RNA expression changes of a few brown markers in eWAT below. Their expression levels are quite low in eWAT and none of them showed significant difference. Since this data doesn't add any new information to the manuscript, we didn't include it in the manuscript.

3. The BAT KO by Ucp1-Cre is a very interesting model. The KO have promoted browning and adipogenesis in BAT and mildly in iWAT, corresponding to the efficiency of Ucp1-Cre. What happens to eWAT? Possibly no phenotype because of no deletion. Then do the mice develop insulin resistance and glucose intolerance as the KO by Adipoq-Cre? This is quite informative to dissect the depot contribution to overall metabolic phenotype, and further support the bifurcation of browning and insulin sensitivity.

It is a very interesting point. We have examined the eWAT but didn't observe any difference in organ weight and marker expression, likely due to the lack of deletion efficiency. We conducted GTT and ITT assay for the Ucp1-Cre KO mice. These animals didn't develop insulin resistance. Thus, the functional influence in BAT by HuR knockout itself is not sufficient to alter the systemic metabolic homeostasis. We have included these data in the revised manuscript (Figure S3C-F).

"As expected, we didn't observe significant difference in fat organ weight and gene expression of most markers (Figure S3C, D). The GTT and ITT assay exhibited little change between the control and HuR-BATKO mice (Figure S3E, F). Thus, the functional influence of HuR knockout in BAT is not sufficient to alter the systemic metabolic homeostasis."

4. The mechanistic studies are solid and insightful. The authors performed RIP-seq to identify targets, then ribosome profiling to exclude the possible effects on translation. Their study on the regulation of Insig-1 by HuR is a strength in this manuscript. It is clearly demonstrated on molecular levels by multiple complementary approaches. However, it is not complete. The author should perform rescue experiment to assure the contribution by Insig-1, such as overexpressing Insig-1 without the nascent 3-UTR in HuR KO, or knockdown Insig-1 in HuR overexpression cells.

As suggested by the reviewer, we have conducted the rescue experiment. This data has been included in our revised manuscript.

"To further examine the functional interactions between HuR and insig1, we overexpressed HuR and knocked down insig1 simultaneously during adipogenesis. Knockdown of Insig1 can indeed attenuate the pro-adipogenic effects from HuR overexpression, indicating that Insig1 is an important downstream target of HuR (Figure S6D)."

5. The findings in the overexpression model of HuR by aP2 promoter are complementary and interesting. There is no effect in chow-fed mice when adipogenesis is at the basal level but KO showed reduced bodyweight gain on HFD feeding when adipogenesis is stimulated. What about inflammation in eWAT? It is expected to decrease according to the KO mice data in Fig. 3. But, if the reduced adipogenesis causes a defect in handling and storing

excessive lipid on HFD feeding, the effects may be neutralized. It will be interesting to examine inflammation in eWAT.

According to the reviewer 1's suggestion, we have removed the transgenic study from this manuscript because this manuscript has been overloaded. We will further characterize the transgenic animals in an independent study.

6. The body composition in Fig. 7B should be presented in absolute weight rather than percentage since their bodyweight is different.

According to the reviewer 1's suggestion, we have removed the transgenic study from this manuscript. We will further characterize the transgenic animals in an independent study.

7. In Fig. 7F: Error bars are missing? The mice are still quite insulin sensitive, meaning the HFD-induced insulin resistance has not been fully established, thus only mild effects on improving insulin sensitivity in the transgenic mice. Same to the insignificant improvement on GTT in Fig. S7J.

According to the reviewer 1's suggestion, we have removed the transgenic study from this manuscript. We will further characterize the transgenic animals in an independent study.

8. The authors should pay more attention to language. Three typos are found in the Abstract: an RNA; inguinal white, and brown adipose tissue; (a) new insight. There are more in the manuscript.

Thanks for pointing them out. We have fixed them.

9. Missing data in Fig 2G.

We have included the data in Fig 2G.

Reviewers' comments:

Reviewer #1 (Remarks to the Author):

The authors have adequately addressed most of my concerns and performed a number of new experiments.

There is only one important issue which needs to be clarified before the manuscript can be published. My former critics concerning the putative role of HMGB1 for reduced myogenesis upon HuR knockdown have not been convincingly addressed by the new Supplementary Figure S8A. The paper by Dormoy-Raclet published in *Nat. Commun.*, 2013, they refer on reports on the inhibitory effect of HuR on HMGB1 translation. This paper showed that HuR binding did not affect the stability of HMGB1 transcripts and consequently, not affect mRNA levels of HMGB1. Therefore, "testing the expression of Hmgb1 in BAT by Real-time PCR" seems not the correct approach to answer this question. Instead, the authors should check for possible changes in HMGB1 protein levels. If yes, their statement "...that downregulation of muscle markers is unlikely due to the down-regulation of HMGB1 but likely due to an enhanced brown adipogenesis" is incorrect and should be changed in accordance.

The same holds true for Figure.S8B demonstrating that the levels of C/EBPbeta are not changed upon HuR knockdown. The paper they refer on (Gantt et al., 2006) demonstrated that HuR target mRNA binding mainly affects the export of bound C/EBPbeta mRNA and this has consequences for its translation. In contrast, the cited paper did not show that HuR depletion has a negative effect on C/EBPb mRNA levels. Corresponding to Fig.. S8A, the authors should show a Western blot for C/EBPbeta.

Reviewer #2 (Remarks to the Author):

The manuscript is significantly improved and all my comments have been addressed.

Reviewer #1 (Remarks to the Author):

The authors have adequately addressed most of my concerns and performed a number of new experiments.

There is only one important issue which needs to be clarified before the manuscript can be published. My former critics concerning the putative role of HMGB1 for reduced myogenesis upon HuR knockdown have not been convincingly addressed by the new Supplementary Figure S8A. The paper by Dormoy-Raclet published in Nat. Commun., 2013, they refer on reports on the inhibitory effect of HuR on HMGB1 translation. This paper showed that HuR binding did not affect the stability of HMGB1 transcripts and consequently, not affect mRNA levels of HMGB1. Therefore, “testing the expression of Hmgb1 in BAT by Real-time PCR” seems not the correct approach to answer this question. Instead, the authors should check for possible changes in HMGB1 protein levels. If yes, their statement “...that downregulation of muscle markers is unlikely due to the down-regulation of HMGB1 but likely due to an enhanced brown adipogenesis” is incorrect and should be changed in accordance. The same holds true for Figure.S8B demonstrating that the levels of C/EBPbeta are not changed upon HuR knockdown. The paper they refer on (Gantt et al., 2006) demonstrated that HuR target mRNA binding mainly affects the export of bound C/EBPbeta mRNA and this has consequences for its translation. In contrast, the cited paper did not show that HuR depletion has a negative effect on C/EBPb mRNA levels. Corresponding to Fig.. S8A, the authors should show a Western blot for C/EBPbeta.

Thanks for the insightful point from this reviewer. We agree that conducting Western blot is necessary to fully address the question whether HuR can target HMGB1 and C/EBPbeta because HuR could influence the translational efficiency of its targets. According to the reviewer's suggestion, we have performed the Western blot to examine the expression of HMGB1 in brown fat and C/EBPbeta in all three adipose tissues. We did not observe any significant difference for these two targets between control and knockout animals. Therefore, HMGB1 and C/EBPbeta are not targeted by HuR in adipose tissue *in vivo*. Our previous statement is not affected. We have included the Western Blot in our revised manuscript (Supplemental Figure 8).

Reviewer #2 (Remarks to the Author):

The manuscript is significantly improved and all my comments have been addressed.

Thanks for this reviewers' positive comments.